# CROSS-MODAL BRAIN GRAPH DIFFUSION

## ABSTRACT

Multimodal brain graph fusion enables the integration of structural and functional information from multiple neuroimaging modalities to advance brain graph analysis. However, existing methods struggle to simultaneously capture (1) intra-modal dependencies (modality-specific topological information) and (2) inter-modal correlations (structural-functional coupling information), both of which are essential attributes specific to multimodal brain graph fusion. This limitation leads to inadequate brain structural-functional information fusion, ultimately failing to correctly reflect the true brain organization. To fill this gap, this paper proposes a novel Cross-modal Brain Graph Diffusion (Xdiff) approach. Xdiff presents a dual graph diffusion mechanism with intra- and inter-modal diffusion modules to capture intra-modal dependencies and inter-modal correlations, respectively. During the diffusion processes, we use an energy constraint function to ensure diffusion consistency, thereby enhancing model stability of learning from multimodal brain graphs. Furthermore, we design a prompt-based fusion strategy to flexibly integrate multimodal features for robust fusion. Empirically, Xdiff achieves state-of-the-art performance on three datasets for brain disorder detection tasks, with accuracy improvements of 4.6%, 2.5%, and 5.6%, respectively[1].

## 1 INTRODUCTION

Along with the rapid advancement of neuroimaging technologies, brain graph analysis has evolved from solely relying on unimodal data to integrating multimodal data for enriched information extraction (Zheng et al., 2022; Qiu et al., 2024; Peng et al., 2024b; Wei et al., 2025). Therefore, multimodal fusion has stood as a prominent trend in various brain graph analysis tasks, such as brain disorder detection and prediction (Cai et al., 2023; Zhang et al., 2024a). Many existing multimodal brain graph fusion studies primarily focus on structural-functional fusion (Popp et al., 2024), where structural brain graphs (SBGs), derived from modalities such as diffusion tensor imaging (DTI), are fused with functional brain graphs (FBGs), typically constructed from functional magnetic resonance imaging (fMRI) (Peng et al., 2025; Yu et al., 2025).

Two essential attributes of structural-functional brain graph fusion are (1) intra-modal dependencies and (2) inter-modal correlations. First, SBGs and FBGs exhibit distinct intra-modal dependencies reflecting their heterogeneous topologies (Lynn & Bassett, 2019). Capturing the specific intra-modal dependencies within each graph can effectively encode modality-specific topological information, which is crucial for multimodal brain graphs. Second, as brain functional and structural connectivity are tightly coupled (Atasoy et al., 2016; Seguin et al., 2023), it is essential to encode the structural-functional coupling information by capturing their inter-modal correlations (Yang et al., 2024). Although existing multimodal brain graph fusion methods have demonstrated significant achievements, they struggle to *simultaneously capture intra-modal dependencies and inter-modal correlations*. This limitation results in inadequate brain structural–functional information fusion, yielding representations that do not correctly reflect brain organization and hamper model performance on various clinical tasks.

In general, fusion methods can be divided into two categories: separate graph representation-based fusion (Ye et al., 2024) and joint graph representation-based fusion (Chen et al., 2022; Cai et al., 2023; Song et al., 2023). Separate graph representation-based fusion methods first learn the representation of each graph independently, preserving their specific intra-modal dependencies, and then

---

[1]The code is available at: https://anonymous.4open.science/r/Xdiff

perform the fusion operation (Figure 1 (a)). For example, RH-BrainFS (Ye et al., 2024) models the specific regional topologies of SBGs and FBGs through subgraph sampling and employs two separate Transformers to learn their representations prior to fusion. Although these methods are effective in retaining intra-modal dependencies, their independent learning mechanism limits the ability to capture inter-modal correlations during the representation learning phase. In contrast, joint graph representation-based fusion methods integrate two graphs into a unified framework to jointly learn multimodal representations, facilitating the extraction of inter-modal correlations (Figure 1 (b)). For instance, Cross-GNN (Yang et al., 2024) enables joint multimodal learning by constructing a unified dynamic graph that integrates both functional and structural features, along with their inter-modal correlations. However, as these methods integrate multimodal graphs into a shared representation space, they can smooth or dilute the modality-specific topology (intra-modal dependency) of each graph.

To fill this gap, this paper proposes a new Cross-modal Brain Graph Diffusion (**Xdiff**) that enables more effective brain structural–functional information fusion by simultaneously capturing intra-modal dependencies and inter-modal correlations (Figure 1 (c)). We first propose a *dual graph diffusion mechanism*, which incorporates both intra-modal diffusion and inter-modal diffusion. These two diffusion processes facilitate intra- and inter-modal feature flow to effectively capture intra-modal dependencies and inter-modal correlations, respectively. Meanwhile, we use an energy constraint function to ensure consistent feature propagation throughout both diffusion processes, thereby enhancing model stability. Then, we design a *prompt-based fusion strategy*, namely PF, leveraging prompt tuning to enable robust multi-modal fusion. Particularly, for each modality, we design a tailored fusion prompt to embed modality-specific cues and flexibly fuse multimodal features.

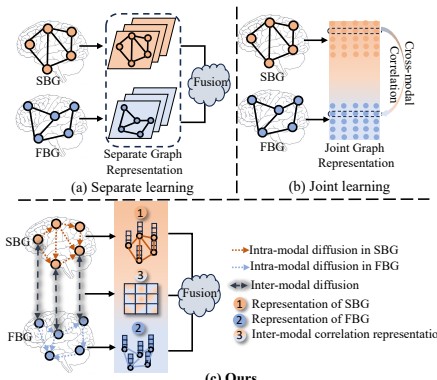

Figure 1: Illustration of different fusion methods.

**Contribution: (1)** We propose a novel cross-modal brain graph diffusion approach, Xdiff, which introduces a dual graph diffusion mechanism to simultaneously capture intra-modal dependencies and inter-modal correlations, enabling more effective brain structural–functional information fusion. **(2)** We develop a prompt-based fusion strategy to flexibly integrate multimodal features, therefore enhancing the robustness of fusion. **(3)** Extensive experimental results show the superiority of our model compared to state-of-the-art methods on various brain disorder detection tasks.

## 2 RELATED WORK

**Multimodal Brain Graph Fusion.** In brain graph analysis, reliance on a single modality often fails to capture features from both brain structure and function, hampering the model's capacity to represent the intricate nature of the brain (Qiu et al., 2024; Qu et al., 2021; Cho et al., 2024b; Xu et al., 2025). Multimodal brain graph fusion, particularly structural-functional brain graph fusion, offers a promising solution, enabling more robust and effective brain data analysis, such as brain disorder detection and prediction (Yang et al., 2023a; Cai et al., 2023; Zhang et al., 2024a). For example, Wen et al. (Wen et al., 2024) analyzed brain age gaps from both brain structure (e.g., grey matter volume and white matter microstructure) and brain functional connectivity to investigate the genetic architecture of brain ageing. RH-BrainFS (Ye et al., 2024) addresses regional heterogeneity in structural and functional brain graphs based on subgraph sampling and Transformer-based fusion bottleneck. MMP-GCN (Song et al., 2023) integrates both fMRI and DTI data while also considering patient demographic and clinical data to construct brain graph topology for early Alzheimer's disease diagnosis. MTAN (Zhu et al., 2022) introduces a self-attention mechanism to extract high-order representations from fMRI and DTI for epilepsy diagnosis. Cross-GNN (Yang et al., 2024) utilizes dynamic brain graphs with mutual learning to capture multimodal associations, extracting inter-modal correlations between brain function and structure. However, these methods face challenges in capturing both intra-modal dependencies and inter-modal correlations of multimodal brain graphs. To address this issue, we propose Xdiff that can simultaneously capture intra-modal depen-

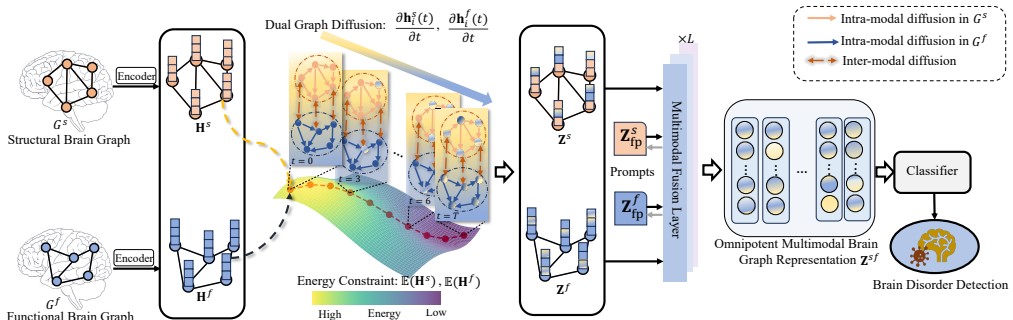

Figure 2: The overall framework of Xdiff.

dencies and inter-modal correlations, enabling more effective brain structural-functional information fusion. More related works about the graph diffusion model and prompt tuning are discussed in Appendix C.

## 3 PRELIMINARIES

**Graph Diffusion.** Given a graph $G = (\mathbf{A}, \mathbf{X})$, following the heat diffusion on a Riemannian manifold, the diffusion process considers all nodes collectively and enables continuous feature flow across nodes. The state (e.g., representation) of node $i$ at time $t$ can be defined as a vector-valued function: $\mathbf{h}_i(t) : [0, \infty] \to \mathbb{R}^d$, where the initial state $\mathbf{h}_i(0) = \mathbf{x}_i \in \mathbf{X}$. The diffusion process describes the evolution of node states through a partial differential equation (PDE) with specified boundary conditions:

$$\frac{\partial \mathbf{H}(t)}{\partial t} = \mathcal{D}(\mathbf{A}(t) \odot \nabla \mathbf{H}(t)). \tag{1}$$

Here, $\mathbf{H}(t) = \{\mathbf{h}_i(t)\}_{i=1}^N$, and $\mathbf{A}(t)$ is the diffusion flow coefficient matrix, which controls the diffusion strength between pairs of nodes at time $t$. Initially, $\mathbf{A}(0) = \mathbf{A}$ is the adjacency matrix of $G$. $\odot$ indicates Hadamard product, and gradient operator $\nabla$ denotes the difference between nodes. $\mathcal{D}$ represent the operator that sums up feature flows at time $t$. For node $i$, the temporal change of heat corresponds to the summation of heat changes across its neighboring nodes in space. Eq. (1) can be explicitly written as:

$$\frac{\partial \mathbf{h}_i(t)}{\partial t} = \sum_{j=1}^N \mathbf{a}_{ij}(t)[\mathbf{h}_j(t) - \mathbf{h}_i(t)], \tag{2}$$

where $\mathbf{a}_{ij}(t) \in \mathbf{A}(t)$, and $N$ is the number of nodes.

## 4 METHODOLOGY

### 4.1 PROBLEM DEFINITION

For a given sample containing an SBG $G^s = (\mathbf{A}^s, \mathbf{X}^s)$ and an FBG $G^f = (\mathbf{A}^f, \mathbf{X}^f)$, the goal of our model is to fuse $G^s$ and $G^f$ while retaining their specific intra-modal dependencies and extracting inter-modal correlations. The feature matrices $\mathbf{X}^s \in \mathbb{R}^{N \times d}$ and $\mathbf{X}^f \in \mathbb{R}^{N \times d}$ represent node features, where $N$ is the node number and $d$ is the feature dimension. Each node is a region of interest (ROI) in the brain. $\mathbf{A}^s \in \mathbb{R}^{N \times N}$ and $\mathbf{A}^f \in \mathbb{R}^{N \times N}$ indicate the adjacency matrices of SBG and FBG, respectively. Notably, the ROIs in the two graphs are defined using the same atlas. Therefore, nodes correspond one-to-one in the two graphs, with the same number of nodes in each. Figure 2 demonstrates the overall framework of Xdiff.

## 4.2 DUAL GRAPH DIFFUSION MECHANISM

### 4.2.1 INTRA- AND INTER-MODAL DIFFUSION

To extend the graph diffusion process to multimodal brain graph scenarios, we design a dual graph diffusion mechanism comprising intra-modal diffusion and inter-modal diffusion. Particularly, the intra-modal diffusion facilitates feature flows within each graph, while the inter-modal diffusion enables feature flows between the two modalities.

Given a sample containing a $G^s = (\mathbf{A}^s, \mathbf{X}^s)$ and a $G^f = (\mathbf{A}^f, \mathbf{X}^f)$, Our dual graph diffusion mechanism reformulates Eq. (2) to accommodate the multimodal scenario. The dual graph diffusion process can be expressed as:

$$\frac{\partial \mathbf{h}_i^s(t)}{\partial t} = \underbrace{\sum_{j=1}^{N} \mathbf{s}_{ij}(t)[\mathbf{h}_j^s(t) - \mathbf{h}_i^s(t)]}_{\text{① intra-modal diffusion in } G^s} + \underbrace{\mathbf{c}_i^s(t)[\mathbf{h}_i^f(t) - \mathbf{h}_i^s(t)]}_{\text{② inter-modal diffusion from } G^s \text{ to } G^f} \quad ,$$

$$\frac{\partial \mathbf{h}_i^f(t)}{\partial t} = \underbrace{\sum_{j=1}^{N} \mathbf{f}_{ij}(t)[\mathbf{h}_j^f(t) - \mathbf{h}_i^f(t)]}_{\text{③ intra-modal diffusion in } G^f} + \underbrace{\mathbf{c}_i^f(t)[\mathbf{h}_i^s(t) - \mathbf{h}_i^f(t)]}_{\text{④ inter-modal diffusion from } G^f \text{ to } G^s} \quad .$$

$$(3)$$

Here, $\mathbf{h}_i^s(0) = \mathbf{x}_i^s \in \mathbf{X}^s$ and $\mathbf{h}_i^f(0) = \mathbf{x}_i^f \in \mathbf{X}^f$. $\mathbf{s}_{ij}(t)$ and $\mathbf{f}_{ij}(t)$ denote the intra-modal diffusion flow coefficients between nodes $i$ and $j$ within graph $G^s$ and graph $G^f$, respectively, at time $t$. The initial value $\mathbf{s}_{ij}(0)$ and $\mathbf{f}_{ij}(0)$ are directly derived from the adjacency matrices of the corresponding graphs. For example, $\mathbf{s}_{ij}(0) = \mathbf{a}_{ij}^s \in \mathbf{A}^s$ and $\mathbf{f}_{ij}(0) = \mathbf{a}_{ij}^f \in \mathbf{A}^f$. $\mathbf{s}_{ij}(t)$ and $\mathbf{f}_{ij}(t)$ dynamically change over time as they are governed by the diffusion process. Meanwhile, $\mathbf{c}_i^s(t)$ represents the inter-modal diffusion flow coefficient of node $i$ from graph $G^s$ to graph $G^f$ at time $t$, while $\mathbf{c}_i^f(t)$ represents the flow coefficient from $G^f$ to $G^s$. $\mathbf{c}_i^s(t)$ and $\mathbf{c}_i^f(t)$ are the correlations between node representations from $G^s$ and $G^f$. Particularly, we employ an attention-based mechanism to calculate these inter-modal diffusion flow coefficients, denoted as: $\mathbf{c}_i^s(t) = \sigma(\mathbf{Q}_i^s(t) \cdot \mathbf{K}_i^f(t))$, $\mathbf{c}_i^f(t) = \sigma(\mathbf{Q}_i^f(t) \cdot \mathbf{K}_i^s(t))$. $\sigma(\cdot)$ is the sigmoid function. $\mathbf{Q}$ and $\mathbf{K}$ represent the query and key matrices, respectively. The values of $\mathbf{c}_i^s(t)$ and $\mathbf{c}_i^f(t)$ depend on the learned node representations and are computed in a dynamic manner. Notably, ① and ③ are intra-modal diffusion processes that effectively capture heterogeneous topologies (e.g., specific intra-modal dependencies) of structural and functional brain graphs, respectively. ② and ④ represent inter-modal diffusion, highlighting inter-modal correlations between brain structure and function.

**Proposition 1.** *In the dual graph diffusion mechanism, the intra-modal diffusion captures the specific intra-modal dependencies within each graph, and the inter-modal diffusion captures the inter-modal correlations.*

The proof of Propostion 1 is given in Appendix B.1. We employ the explicit Euler method, a numerical technique for approximating solutions to ordinary differential equations (ODEs), to solve the continuous dynamics described in Eq. (3). With diffusion step size $\rho$, the explicit Euler method is as:

$$\mathbf{h}_i^{(s,k+1)} = \underbrace{(1-2\rho)\mathbf{h}_i^{(s,k)} + \rho \sum_j^N \mathbf{s}_{ij}^k \mathbf{h}_i^{(s,k)}}_{\text{intra-modal}} + \underbrace{\rho \mathbf{c}_i^{(s,k)} \mathbf{h}_i^{(f,k)}}_{\text{inter-modal}},$$

$$\mathbf{h}_i^{(f,k+1)} = \underbrace{(1-2\rho)\mathbf{h}_i^{(f,k)} + \rho \sum_j^N \mathbf{f}_{ij}^k \mathbf{h}_i^{(f,k)}}_{\text{intra-modal}} + \underbrace{\rho \mathbf{c}_i^{(f,k)} \mathbf{h}_i^{(s,k)}}_{\text{inter-modal}}.$$

$$(4)$$

Here, $k$ indicates the $k$-th diffusion layer, $\mathbf{h}_i^{(s,0)} = \mathbf{x}_i^s \in \mathbf{X}^s$ and $\mathbf{h}_i^{(f,0)} = \mathbf{x}_i^f \in \mathbf{X}^f$.

**Theorem 1.** *During the iterative convergence of Eq.(4), the diffusion step size $\rho$ satisfies the condition: $0 < \rho < 1$.*

The proof is given in Appendix B.2.

### 4.2.2 ENERGY CONSTRAINT

During the diffusion process, defining appropriate intra-modal and inter-modal diffusion flow coefficients is essential to maximize information utility and ensure diffusion consistency. Inspired by (Wu et al., 2023a), we use an energy constraint function to constrain the diffusion process, thereby promoting consistent diffusion and consequently enhancing model stability. The energy constraint function is formulated as follows:

$$\mathbb{E}(\mathbf{H}^s) = w_1 \underbrace{\sum_{i,j}^{N} \beta(\|\mathbf{h}_i^s - \mathbf{h}_j^s\|_2^2)}_{\text{\textcircled{1}}} + \underbrace{w_2\alpha(\|\mathbf{H}^f - \mathbf{H}^s\|_{\mathcal{F}}^2)}_{\text{\textcircled{2}}},$$

$$\mathbb{E}(\mathbf{H}^f) = w_1 \underbrace{\sum_{i,j}^{N} \beta(\|\mathbf{h}_i^f - \mathbf{h}_j^f\|_2^2)}_{\text{\textcircled{3}}} + \underbrace{w_2\alpha\|\mathbf{H}^s - \mathbf{H}^f\|_{\mathcal{F}}^2}_{\text{\textcircled{4}}}.$$

$$(5)$$

Here, $\|\cdot\|_2$ stands as the Euclidean norm of vectors, and $\|\cdot\|_{\mathcal{F}}$ denotes the Frobenius norm of matrices. $\beta(\cdot)$ and $\alpha(\cdot)$ are non-decreasing concave functions. $w_1$ and $w_2$ are weighting constants. ⚠️1 and 🔺3 quantify the difference between nodes within each modality, while 🔺2 and 🔺4 quantify the difference across two modalities. Thus, lower difference (both intra-modal and inter-modal differences) leads to lower energy. Node representations will evolve to produce lower energy, and the final representation is expected to exhibit the lowest energy. The minimization of the energy function can be converted to a minimization of its variational upper bound, which is formulated as the following proposition.

**Proposition 2.** *The upper bound of the energy constraint function (Eq. (5)) is:*

$$\tilde{\mathbb{E}}(\mathbf{H}^s) = w_1 \sum_{i,j}^{N} \left( \mathbf{s}_{ij}\|\mathbf{h}_i^s - \mathbf{h}_j^s\|_2^2 - \tilde{\beta}(\mathbf{s}_{ij}) \right) + w_2 \left( \mathbf{C}^s\|\mathbf{H}^f - \mathbf{H}^s\|_{\mathcal{F}}^2 - \tilde{\alpha}(\mathbf{C}^s) \right),$$

$$\tilde{\mathbb{E}}(\mathbf{H}^f) = w_1 \sum_{i,j}^{N} \left( \mathbf{f}_{ij}\|\mathbf{h}_i^f - \mathbf{h}_j^f\|_2^2 - \tilde{\beta}(\mathbf{f}_{ij}) \right) + w_2 \left( \mathbf{C}^f\|\mathbf{H}^s - \mathbf{H}^f\|_{\mathcal{F}}^2 - \tilde{\alpha}(\mathbf{C}^f) \right).$$

$$(6)$$

*Here, $\mathbf{C}^s = \{\mathbf{c}_i^s\}_{i=1}^N$ and $\mathbf{C}^f = \{\mathbf{c}_i^f\}_{i=1}^N$. $\tilde{\beta}(\cdot)$ and $\tilde{\alpha}(\cdot)$ are the concave conjugate functions of $\beta(\cdot)$ and $\alpha(\cdot)$, respectively. This upper bound is achieved if and only if $\mathbf{s}_{ij}$, $\mathbf{f}_{ij}$, $\mathbf{C}^s$, and $\mathbf{C}^f$ satisfy:*

$$\mathbf{s}_{ij} = \left.\frac{\partial\beta(\mathbf{p}^2)}{\partial\mathbf{p}^2}\right|_{\mathbf{p}=\|\mathbf{h}_i^s - \mathbf{h}_j^s\|_2^2}, \mathbf{f}_{ij} = \left.\frac{\partial\beta(\mathbf{q}^2)}{\partial\mathbf{q}^2}\right|_{\mathbf{q}=\|\mathbf{h}_i^f - \mathbf{h}_j^f\|_2^2},$$

$$\mathbf{C}^s = \mathbf{C}^f = \left.\frac{\partial\beta(\mathbf{O}^2)}{\partial\mathbf{O}^2}\right|_{\mathbf{O}=\|\mathbf{H}^f - \mathbf{H}^s\|_{\mathcal{F}}^2 = \|\mathbf{H}^s - \mathbf{H}^f\|_{\mathcal{F}}^2}.$$

$$(7)$$

The proof is provided in Appendix B.3. With energy constraint, Eq. (4) can be rewritten as:

$$\mathbf{h}_i^{(s,k+1)} = (1 - 2\rho)\mathbf{h}_i^{(s,k)} + \rho\sum_j^{N} \mathbf{s}_{ij}^k\mathbf{h}_i^{(s,k)} + \rho\mathbf{c}_i^{(s,k)}\mathbf{h}_i^{(f,k)},$$

$$\mathbf{h}_i^{(f,k+1)} = (1 - 2\rho)\mathbf{h}_i^{(f,k)} + \rho\sum_j^{N} \mathbf{f}_{ij}^k\mathbf{h}_i^{(f,k)} + \rho\mathbf{c}_i^{(f,k)}\mathbf{h}_i^{(s,k)}.$$

$$(8)$$

$$\text{s.t. } \mathbb{E}(\mathbf{H}^{(s,k)}) \geq \mathbb{E}(\mathbf{H}^{(s,k+1)}), \mathbb{E}(\mathbf{H}^{(f,k)}) \geq \mathbb{E}(\mathbf{H}^{(f,k+1)}).$$

Consequently, after the diffusion mechanism, for node $i$, we obtain its representations $\mathbf{z}_i^s = \mathbf{h}_i^{(s,K)}$ and $\mathbf{z}_i^f = \mathbf{h}_i^{(f,K)}$ for structural brain graph and functional brain graph, respectively. $K$ is the number of diffusion layers. The output representations capture both intra-modal dependencies and inter-model correlations.

### 4.3 PROMPT-BASED FUSION STRATEGY

We then develop a prompt-based fusion (PF) strategy to integrate the obtained multimodal representations of structural and functional brain graphs. Because prompts can flexibly guide the model to focus on specific features of each modality, PF can achieve robust multimodal fusion. Particularly, we leverage fusion prompts that embed modality-specific cues to guide the model in extracting the most informative features from each modality, enabling effective and robust multimodal fusion while preserving the unique information of each modality.

To implement this, we define two distinct fusion prompts for the structural and functional brain graphs, denoted as $\mathbf{Z}_{\mathrm{fp}}^s \in \mathbb{R}^U$ and $\mathbf{Z}_{\mathrm{fp}}^f \in \mathbb{R}^U$, respectively. $U$ indicates the prompt length. These prompts are then concatenated with their corresponding input representations to enhance the information flow from each modality:

$$
\begin{aligned}
\hat{\mathbf{Z}}^s &= \mathbf{Z}^s \oplus \mathbf{Z}_{\mathrm{fp}}^s, \\
\hat{\mathbf{Z}}^f &= \mathbf{Z}^f \oplus \mathbf{Z}_{\mathrm{fp}}^f.
\end{aligned}
\tag{9}
$$

Here, $\mathbf{Z}^s = \{\mathbf{z}_i^s\}_{i=1}^N$ and $\mathbf{Z}^f = \{\mathbf{z}_i^f\}_{i=1}^N$ represent the node features in structural and functional modalities, respectively. $\oplus$ is the concatenation operation. By appending the prompts $\mathbf{Z}_{\mathrm{fp}}^s$ and $\mathbf{Z}_{\mathrm{fp}}^f$, we equip each modality with enriched context, guiding the subsequent fusion step. In the fusion module, the concatenated representations $\hat{\mathbf{Z}}^s$ and $\hat{\mathbf{Z}}^f$ are fused to generate the multimodal representation $\mathbf{Z}^{sf}$. At each layer, this fusion is expressed as:

$$
\mathbf{Z}^{(sf,l+1)} = \hat{\mathbf{Z}}^{(s,l)} \oplus \hat{\mathbf{Z}}^{(f,l)},
\tag{10}
$$

where $l$ indicates the $l$-th layer. The number of fusion layers is $L$. Afterwards, we apply the cross-entropy loss function for classification tasks:

$$
\mathcal{L} = -\frac{1}{V} \sum_{v=1}^{V} \sum_{m=1}^{M} y_{v,m} \log \left( \mathrm{softmax}(\mathbf{Z}_v^{sf})_m \right),
\tag{11}
$$

where $V$ and $M$ denote the number of samples and classes, respectively. $\mathbf{Z}_v^{sf}$ is the multimodal representation of sample $v$, and $y_{v,m}$ is the true label indicator for sample $v$.

## 5 EXPERIMENTS

### 5.1 EXPERIMENTAL SETUP

**Datasets.** We construct structural brain graphs by measuring cosine similarity between ROIs derived from DTI data, while functional brain graphs are constructed by computing the Pearson correlation coefficient (PCC) between ROIs of fMRI data. We conduct experiments on three benchmark datasets. (1) Alzheimer's Disease Neuroimaging Initiative (ADNI) [2] dataset contains fMRI and DTI data of 407 subjects, including 190 normal controls (NCs), 170 mild cognitive impairment (MCI) patients, and 47 Alzheimer's disease (AD) patients. (2) Parkinson's Progression Markers Initiative (PPMI) dataset (Marek et al., 2018) includes fMRI and DTI data of 49 Parkinson's disease (PD) patients, 69 individuals at risk for PD (Prodromal), and 40 NCs. (3) 4-Repeat Tauopathy Neuroimaging Initiative (4RTNI) dataset [3] contains fMRI and DTI data of 31 samples of progressive supranuclear palsy (PSP) and 47 samples of corticobasal syndrome (CBS). The ROI definition for both DTI and fMRI data in three datasets is based on the automated anatomical labeling (AAL) atlas (Tzourio-Mazoyer et al., 2002).

---

[2]https://adni.loni.usc.edu/
[3]http://memory.ucsf.edu/research/studies/4rtni

**Baselines.** We compare our model with state-of-the-art baselines. We first select five representative unimodal brain graph learning models for comparison, including, A-GCL (Zhang et al., 2023), STAGIN (Kim et al., 2021), GroupBNA (Peng et al., 2024a), MCST-GCN (Zhu et al., 2024), NeuroPath (Wei et al., 2024). The modality used remains the same as that of the original models. To demonstrate the advantages of our model in multimodal scenarios, we then compare our model with representative multimodal brain graph learning models, including Cross-GNN (Yang et al., 2024), RH-BrainFS (Ye et al., 2024), MTAN (Zhu et al., 2022), and AL-NEGAT (Chen et al., 2022). In addition, we compare our method with three representative graph diffusion models, Difformer (Wu et al., 2023b), DDM (Yang et al., 2023b), and ECMGD (Lu et al., 2024), and two typical machine learning approaches, Support Vector Machine (SVM) and Random Forest (RF), in multimodal scenarios. For each sample, these methods take the concatenation of the structural brain graph and the functional brain graph as input. All baselines are executed using their optimal configurations.

**Evaluation Metrics.** We evaluate our model on graph classification tasks, where the ADNI and PPMI datasets involve multiclass classification problems, while the 4RTNI dataset corresponds to a binary classification problem. Five metrics are used to evaluate the model performance, including test accuracy (ACC), F1 score, area under the receiver operating characteristic curve (AUC), sensitivity (Sen.), and specificity (Spe.). Particularly, for the multiclass classification tasks, we use macro averaging for the F1 score, Sen., and Spe. All results are the average values of 5 random runs on test sets with the standard deviation.

**Implementation Details.** Our model is implemented using PyTorch v2.2.0. Model training is performed on an NVIDIA 3090 GPU with 24GB of memory. All datasets are randomly split into 70% for training and 10% used for validation, and 20% for testing. More implementation details and settings are given in Appendix D.

## 5.2 RESULTS

**Model Performance Comparison.** Table 1 compares the results (ACC and AUC) of Xdiff with baselines. The results for F1, Sen., and Spe. are provided in Table 4 (see Appendix E.1). As shown by the experimental results, Xdiff consistently outperforms other baselines across all three datasets. Notably, Xdiff achieves the highest accuracy of 70.8%, 62.5%, and 77.8% on three datasets, respectively, representing improvements of 4.6%, 2.5%, and 5.6% over the second-best method. The experimental results demonstrate that our model excels in various brain disorder detection tasks.

Table 1: Experimental results (ACC and AUC) on three datasets (%). The best results are marked in bold, and the suboptimal results are marked underlined. $\Delta_{SOTA}$ indicates the improvements or reductions of Xdiff compared to SOTA methods.

| Method | | ADNI | | PPMI | | 4RTNI | |
|---|---|---|---|---|---|---|---|
| | | ACC | AUC | ACC | AUC | ACC | AUC |
| Unimodal | A-GCL | 51.7±4.8 | 61.5±7.9 | 43.9±7.5 | 60.9±9.1 | 58.2±12.2 | 54.1±9.8 |
| | STAGIN | 61.7±2.5 | 72.0±3.3 | 56.3±5.2 | **67.5±4.5** | 71.0±10.1 | 61.4±13.1 |
| | GroupBNA | 49.6±1.5 | 52.2±2.4 | 46.7±4.4 | 61.3±2.8 | 53.3±3.9 | 52.8±5.4 |
| | MCST-GCN | 54.1±3.6 | 67.8±3.7 | 53.1±5.4 | 64.9±6.0 | 66.7±8.5 | 61.4±15.3 |
| | NeuroPath | 60.9±4.3 | 73.1±3.5 | 57.3±4.6 | 58.0±5.4 | 70.5±10.2 | 64.1±7.4 |
| Multimodal | RF | 52.0±6.3 | 65.2±5.1 | 48.4±5.4 | 61.5±3.0 | 58.8±4.7 | 53.8±10.8 |
| | SVM | 55.1±7.3 | 67.3±4.4 | 44.1±4.1 | 57.1±3.3 | 58.0±9.8 | 54.9±10.9 |
| | Difformer | 59.0±4.3 | 68.9±4.0 | 49.0±8.4 | 59.4±10.5 | 61.0±10.8 | 53.1±12.9 |
| | DDM | 58.5±3.7 | 76.2±3.1 | 53.7±12.2 | 61.8±9.3 | 60.5±13.9 | 45.1±19.7 |
| | ECMGD | 66.2±1.1 | 70.1±1.8 | 57.0±3.3 | 60.8±5.2 | 71.3±5.4 | 68.6±7.1 |
| | Cross-GNN | 55.8±5.5 | 62.6±4.5 | 58.0±3.4 | 65.9±7.5 | 72.2±5.6 | 72.2±12.5 |
| | RH-BrainFS | 53.7±9.1 | 70.1±6.4 | 48.6±8.0 | 66.3±9.2 | 71.6±12.3 | 69.5±14.3 |
| | MTAN | 59.0±6.8 | 49.8±6.2 | 55.4±8.0 | 42.4±7.1 | 62.5±22.4 | 56.0±15.1 |
| | AL-NEGAT | 63.4±2.8 | 72.8±3.3 | 60.0±4.5 | 65.8±6.7 | 70.0±10.0 | 66.7±20.2 |
| Our Model | **Xdiff** | **70.8±0.6** | **77.2±0.9** | **62.5±0.1** | 64.7±1.2 | **77.8±3.1** | **77.4±0.4** |
| | $\Delta_{SOTA}$ | ↑ 4.6 | ↑ 1.0 | ↑ 2.5 | ↓ 2.8 | ↑ 5.6 | ↑ 5.2 |

**Stability Analysis.** As shown in Table 1 and Table 4, Xdiff exhibits lower standard deviation values than other baselines across five evaluation metrics on three datasets, demonstrating its enhanced stability. To further validate this, we compare the 95% confidence intervals of accuracy between Xdiff and other four multimodal brain graph learning models on three datasets, as shown in Fig-

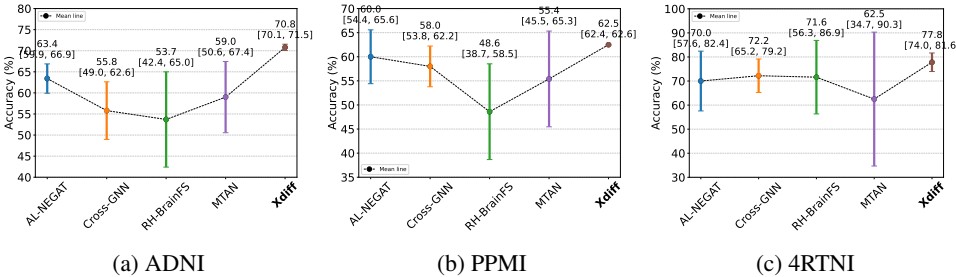

(a) ADNI  (b) PPMI  (c) 4RTNI

Figure 3: Comparison of the 95% confidence intervals of accuracy across different methods on three datasets.

ure 3. Notably, our model is significantly more stable compared to other models, with the accuracy confidence intervals ranging from 70.1% to 71.5% on ADNI, 62.4% to 62.6% on PPMI, and 74.0% to 81.6% on 4RTNI.

## 5.3 ABLATION STUDY

**Effectiveness of Dual Graph Diffusion.** To verify how our dual graph diffusion mechanism benefits the model performance, we conduct various ablation experiments by (1) removing intra-modal diffusion, denoted as "w/o Intra"; (2) removing inter-modal diffusion, denoted as "w/o Inter"; (3) removing both intra- and inter-modal diffusion, denoted as "w/o Intra & Inter".

Table 2: Model performance (ACC and AUC) of Xdiff and its variants on three datasets (%).

| Method | ADNI | | PPMI | | 4RTNI | |
|---|---|---|---|---|---|---|
| | ACC | AUC | ACC | AUC | ACC | AUC |
| w/o Intra | 63.6±0.6 | 71.2±0.1 | 56.3±0.1 | 62.6±0.2 | 60.0±0.5 | 70.4±1.0 |
| w/o Inter | 68.5±1.9 | 71.7±0.1 | 61.5±1.5 | 62.9±0.1 | 73.3±0.5 | 67.7±1.1 |
| w/o Intra & Inter | 63.6±0.6 | 70.7±0.2 | 56.3±0.2 | 62.2±0.2 | 62.2±3.1 | 69.3±0.8 |
| **Xdiff** | **70.8±0.6** | **77.2±0.9** | **62.5±0.1** | **64.7±1.2** | **77.8±3.1** | **77.4±0.4** |

The results on three datasets are summarized in Table 2 and Table 5 (see Appendix E.2). The overall performance of Xdiff is significantly better than that of other variants. This indicates that our dual graph diffusion mechanism is crucial for the model performance, suggesting both intra-modal dependencies and inter-modal correlations are essential for more effective brain structural-functional information fusion.

**Effectiveness of Energy Constraint Function.** As discussed in Section 5.2, Xdiff demonstrates greater stability compared to other methods. We claim that the energy constraint function plays a critical role in improving model stability. To verify this, we compare the 95% confidence intervals of accuracy between Xdiff and its variant without the energy constraint function, denoted as "w/o Energy". The results are shown in Figure 4. The results on three datasets reveal that the model becomes less stable when the energy constraint is removed. This demonstrates the effectiveness of the energy constraint in enhancing model stability and highlights its critical role in ensuring consistent feature propagation throughout the diffusion processes. More detailed analysis is provided in Appendix E.2.

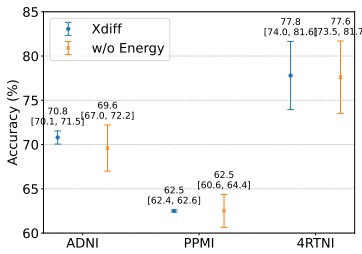

Figure 4: Comparison of the 95% confidence intervals of accuracy between Xdiff and its variant.

**Effectiveness of Prompt-based Fusion.** To evaluate the effectiveness of our prompt-based fusion strategy, we replace it with other fusion operations, including Hadamard multiplication (Multi.), summation (Sum.), concatenation (Concat.), attention-based fusion (Atten.), and Transformer-based fusion (Trans.). In addition, to verify that the fusion prompts can enhance the robustness of multimodal fusion, we introduce Gaussian noise (noise ratio = 0.4) into $\mathbf{Z}^s$ and $\mathbf{Z}^f$ before fusion. We then compare the performance reduction of models employing different fusion methods to evaluate the noise resilience of each approach. Figure 5 illustrates the accuracy changes of models using different fusion methods before and after noise interference on three datasets. Detailed results for all five evaluation metrics on three datasets are shown in Table 6, Table 7, and Table 8 (see Appendix E.2). As the results indicate, Xdiff with PF strategy outperforms models using the other three fusion methods. Furthermore, Xdiff with PF strategy demonstrates relatively stable performance, indicating that our PF strategy effectively enhances model robustness

against noise interference. For example, the accuracy on the ADNI dataset remains unaffected in the presence of noise interference. This is because prompts can flexibly guide the model to focus on specific features of each modality, thereby effectively fusing multimodal features. More explanation is provided in Appendix E.2.

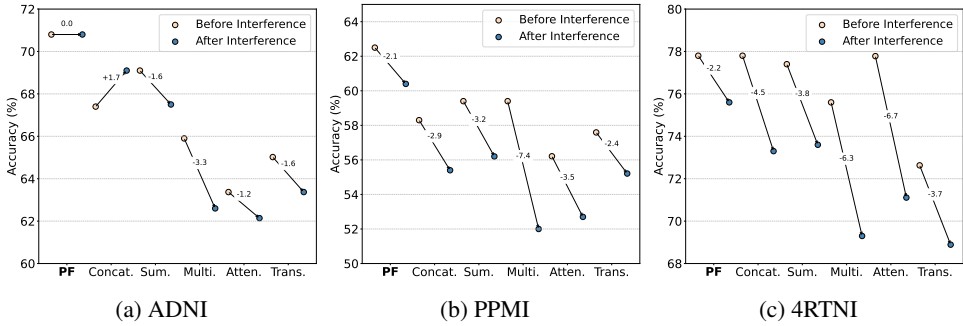

(a) ADNI            (b) PPMI            (c) 4RTNI

Figure 5: Accuracy changes of models using different fusion methods before and after noise interference on three datasets.

## 5.4 HYPERPARAMETER STUDY

**Impact of $\rho$ and $K$ in Diffusion Process.** We explore the impact of diffusion layer number $K$ and step size $\rho$ on model performance. We conduct experiments while keeping all other parameters unchanged. Figure 6 (a) shows the experimental results (accuracy) on the ADNI dataset, while results on PPMI and 4RTNI are presented in Figure 8 (see Appendix E.3). We can see that the model generally outperforms the SOTA method when $K$ is 1, 2, or 4. A notable decline in performance is observed as $K = 8$. This may be due to issues such as over-smoothing, noise accumulation, and gradient vanishing. Notably, when $K$ is set to a relatively small value, the model performance remains superior across all values of $\rho$.

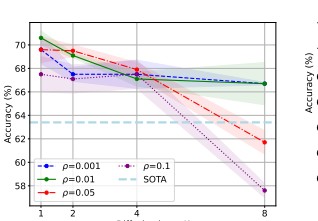

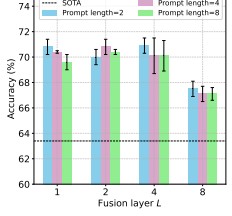

(a) Performance of Xdiff w.r.t $\rho$ and $K$ on ADNI.

(b) Performance of Xdiff w.r.t $L$ and $U$ on ADNI.

Figure 6: Results of hyperparameter study.

**Impact of Fusion Layer Number and Prompt Length.** We investigate the parameter sensitivity of Xdiff by examining the impact of fusion layer number $L$ and prompt length $U$. Figure 6 (b) illustrates the results of accuracy on the ADNI dataset, while results on PPMI and 4RTNI are shown in Figure 9 (see Appendix E.3). The results demonstrate that the model performance remains relatively stable, with only slight variations observed across different prompt lengths (2, 4, or 8) and fusion layer numbers (1, 2, 4, and 8). Overall, Xdiff with different parameter settings generally outperforms the SOTA method. This indicates the adaptability of the proposed model to varying parameter configurations.

## 6 CONLUSION

This paper proposes a cross-modal brain graph diffusion (Xdiff) to simultaneously capture intra-modal dependencies and inter-modal correlations, achieving more effective brain structural-functional information fusion. Extensive experiments on three datasets demonstrate that Xdiff not only outperforms state-of-the-art baselines but also exhibits superior model stability and robustness. Xdiff provides valuable insights into improving the efficacy of multimodal brain graph fusion, which can greatly benefit a wide range of brain graph analysis tasks. However, due to the limited available multimodal brain imaging datasets, the application of Xdiff to broader tasks is constrained. More detailed limitations and potential impacts are discussed in Appendix F.

ETHICS STATEMENT

This work adheres to the ICLR Code of Ethics. This work uses publicly available neuroimaging datasets (ADNI, PPMI, and 4RTNI), all of which were collected under established ethical guidelines with informed consent from participants and approval from relevant institutional review boards. We strictly followed the data usage agreements and ensured that no personally identifiable information is included in our analysis. Our study focuses on methodological contributions in brain graph learning. The potential societal benefit is to advance reliable tools for diagnosis of neurological diseases. However, we acknowledge that any automated system in healthcare carries risks of misuse or over-interpretation. To mitigate such risks, we emphasize that our method should be regarded as a research tool, not a clinical diagnostic system. Further clinical validation is required before deployment. We declare that there are no conflicts of interest, sponsorship biases, or ethical concerns regarding data privacy, fairness, or security.

REPRODUCIBILITY STATEMENT

We have made the best effort to ensure reproducibility. We have provided a link to an anonymous downloadable source code. Detailed experimental setup has been described in Appendix D.

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

## A  THE USE OF LARGE LANGUAGE MODELS

Large language models (LLMs) are solely used to aid and polish writing in this manuscript. No data, results, analysis or conclusions are generated by LLMs.

## B  THEORETICAL ANALYSIS

### B.1  PROOF FOR PROPOSITION 1

**Proposition 1.** *In the dual graph diffusion mechanism, the intra-modal diffusion captures the specific intra-modal dependencies within each graph, and the inter-modal diffusion captures the inter-modal correlations.*

*Proof.* According to the theory of graph diffusion processes, the change in the feature of node $i$ is governed by the summation of feature fluxes entering $i$ from other connected nodes in the graph, reflecting the interactions between nodes in the spatial domain (Chamberlain et al., 2021). Therefore, for intra-modal diffusion in $G^s$, term $\sum_{j=1}^{N} \mathbf{s}_{ij}(t)[\mathbf{h}_j^s(t) - \mathbf{h}_i^s(t)]$ captures the summation of feature fluxes entering node $i$ from other nodes within the graph $G^s$. This term effectively models the influence of neighboring nodes within $G^s$, capturing spatial dependencies and the unique topology of the structural brain graph. Similarly, the intra-modal diffusion in $G^f$, represented by $\sum_{j=1}^{N} \mathbf{f}_{ij}(t)[\mathbf{h}_j^f(t) - \mathbf{h}_i^f(t)]$, captures the unique topology associated with the functional brain graph. For inter-modal diffusion, the terms $\mathbf{c}_i^s(t)[\mathbf{h}_i^f(t) - \mathbf{h}_i^s(t)]$ and $\mathbf{c}_i^f(t)[\mathbf{h}_i^s(t) - \mathbf{h}_i^f(t)]$ represent the feature fluxes exchanged between two modalities. These terms capture the rate of feature transfer from one modality to another at node $i$, modeling the inter-modal correlations. □

### B.2  PROOF FOR THEOREM 1

**Theorem 1.** *During the iterative convergence of Eq.(4), the diffusion step size $\rho$ satisfies the condition:* $0 < \rho < 1$.

*Proof.* Let us consider the structural brain graph $G^s$ as an example, the first line of Eq.(4)

$$\mathbf{h}_i^{(s,k+1)} = (1 - 2\rho)\mathbf{h}_i^{(s,k)} + \rho \sum_j^N \mathbf{s}_{ij}^k \mathbf{h}_i^{(s,k)} + \rho \mathbf{c}_i^{(s,k)} \mathbf{h}_i^{(f,k)}$$

can be rewritten in matrix form as

$$\mathbf{H}^{(s,k+1)} = \mathbf{M}^k \mathbf{H}^{(s,k)} + \rho \mathbf{C}^{(s,k)} \mathbf{H}^{(f,k)}.$$

Here,

$$\mathbf{M}^k = (1 - 2\rho)\mathbf{I} + \rho \mathbf{S}^k,$$

where, $\mathbf{H}^{(s,k)} = \{\mathbf{h}_i^{(s,k)}\}_{i=1}^N$, $\mathbf{C}^{(s,k)} = \{\mathbf{c}_i^{(s,k)}\}_{i=1}^N$, and $\mathbf{S}^k = \{\mathbf{s}_{ij}^k\}_{i,j=1}^N$. Assume $\mathbf{S}^k$ is a stochastic matrix, which implies that its largest eigenvalue is $\lambda_{\max}(\mathbf{S}^k) = 1$. For the iterative process to converge, the spectral radius of $\mathbf{M}^k$ must be less than 1, leading to the convergence condition

$$|(1 - 2\rho) + \rho\lambda_{\max}| < 1.$$

Substituting $\lambda_{\max} = 1$ from our assumption on $\mathbf{S}^k$, we have

$$|1 - 2\rho + \rho| < 1.$$

This simplifies further to

$$|1 - \rho| < 1,$$

which results in the inequality

$$0 < \rho < 1.$$

Therefore, the diffusion step size $\rho$ must lie strictly between 0 and 1 for convergence. This condition also holds when considering functional brain graph $G^f$. □

### B.3 PROOF FOR PROPOSITION 2

**Proposition 2.** *The upper bound of the energy constraint function (Eq. (5)) is:*

$$\tilde{\mathbb{E}}(\mathbf{H}^s) = w_1 \sum_{i,j}^{N} \left( \mathbf{s}_{ij} \| \mathbf{h}_i^s - \mathbf{h}_j^s \|_2^2 - \tilde{\beta}(\mathbf{s}_{ij}) \right) + w_2 \left( \mathbf{C}^s \| \mathbf{H}^f - \mathbf{H}^s \|_{\mathcal{F}}^2 - \tilde{\alpha}(\mathbf{C}^s) \right),$$

$$\tilde{\mathbb{E}}(\mathbf{H}^f) = w_1 \sum_{i,j}^{N} \left( \mathbf{f}_{ij} \| \mathbf{h}_i^f - \mathbf{h}_j^f \|_2^2 - \tilde{\beta}(\mathbf{f}_{ij}) \right) + w_2 \left( \mathbf{C}^f \| \mathbf{H}^s - \mathbf{H}^f \|_{\mathcal{F}}^2 - \tilde{\alpha}(\mathbf{C}^f) \right).$$

*Here, $\mathbf{C}^s = \{\mathbf{c}_i^s\}_{i=1}^N$ and $\mathbf{C}^f = \{\mathbf{c}_i^f\}_{i=1}^N$. $\tilde{\beta}(\cdot)$ and $\tilde{\alpha}(\cdot)$ are the concave conjugate functions of $\beta(\cdot)$ and $\alpha(\cdot)$, respectively. This upper bound is achieved if and only if $\mathbf{s}_{ij}$, $\mathbf{f}_{ij}$, $\mathbf{C}^s$, and $\mathbf{C}^f$ satisfy:*

$$\mathbf{s}_{ij} = \frac{\partial \beta(\mathbf{p}^2)}{\partial \mathbf{p}^2} \bigg|_{\mathbf{p} = \| \mathbf{h}_i^s - \mathbf{h}_j^s \|_2^2}, \mathbf{f}_{ij} = \frac{\partial \beta(\mathbf{q}^2)}{\partial \mathbf{q}^2} \bigg|_{\mathbf{q} = \| \mathbf{h}_i^f - \mathbf{h}_j^f \|_2^2}, \mathbf{C}^s = \mathbf{C}^f = \frac{\partial \beta(\mathbf{O}^2)}{\partial \mathbf{O}^2} \bigg|_{\mathbf{O} = \| \mathbf{H}^f - \mathbf{H}^s \|_{\mathcal{F}}^2 = \| \mathbf{H}^s - \mathbf{H}^f \|_{\mathcal{F}}^2}.$$

*Proof.* We specifically apply the variational form of concave functions to derive the upper bound. According to Fenchel duality, any non-decreasing concave function $\gamma(\cdot)$ can be expressed as a variational bound, leading to a decomposition. Therefore, we obtain the following expressions

$$\gamma(\mathbf{p}^2) = \min_{\mu \geq 0} [\mu \mathbf{p}^2 - \tilde{\gamma}(\mu)] \geq \mu \mathbf{p}^2 - \tilde{\gamma}(\mu),$$

$$\gamma(\mathbf{q}^2) = \min_{\mu \geq 0} [\mu \mathbf{q}^2 - \tilde{\gamma}(\mu)] \geq \mu \mathbf{q}^2 - \tilde{\gamma}(\mu),$$

$$\gamma(\mathbf{O}^2) = \min_{\mu \geq 0} [\mu \mathbf{O}^2 - \tilde{\gamma}(\mu)] \geq \mu \mathbf{O}^2 - \tilde{\gamma}(\mu).$$

Here, $\mu$ stands as a variational parameter, and $\tilde{\gamma}(\cdot)$ is the concave conjugate function of $\gamma(\cdot)$. These bounds define $\gamma(\mathbf{p}^2)$, $\gamma(\mathbf{q}^2)$, and $\gamma(\mathbf{O}^2)$ as the minimal envelope of quadratic bounds, parameterized by $\mu \geq 0$. By substituting each term in Eq. (5) with these bounds, we obtain the upper bound given in Eq. (6). The necessary and sufficient condition for equality to hold in this upper bound is as follows

$$\ddot{\mu} \mathbf{p}^2 - \tilde{\gamma}(\ddot{\mu}) = \gamma(\mathbf{p}^2), \text{where} \quad \ddot{\mu} = \frac{\partial \beta(\mathbf{p}^2)}{\partial \mathbf{p}^2},$$

$$\breve{\mu} \mathbf{q}^2 - \tilde{\gamma}(\breve{\mu}) = \gamma(\mathbf{q}^2), \text{where} \quad \breve{\mu} = \frac{\partial \beta(\mathbf{q}^2)}{\partial \mathbf{q}^2},$$

$$\hat{\mu} \mathbf{O}^2 - \tilde{\gamma}(\hat{\mu}) = \gamma(\mathbf{O}^2), \text{where} \quad \hat{\mu} = \frac{\partial \beta(\mathbf{O}^2)}{\partial \mathbf{O}^2}.$$

□

## C RELATED WORK

**Graph Diffusion Model.** The graph diffusion process is governed by partial differential equations (PDEs), which model the spread of information or features across the graph structure over time (Chamberlain et al., 2021; Thorpe et al., 2022; Wu et al., 2023a; Chopin et al., 2024; Zhang et al., 2024c; Bamberger et al., 2025; Li et al., 2025; Lu et al., 2024). Song et al. (Song et al., 2022) proposed a graph neural PDE framework that generalizes heat flow into broader flow schemes, enhancing the robustness of graph neural networks (GNNs). NDM (Song et al., 2022) utilizes a unique diffusion process to capture the distinct features of each node and provides a generalized heat diffusion function that models varying diffusion patterns across graphs. Wave-GD (Cho et al., 2024a) introduces a multi-resolution diffusion strategy leveraging spectral coherence to enhance graph diffusion. HiD-Net (Li et al., 2024b) incorporates the diffusion equation with the fidelity term, establishing connections between the diffusion process and various GNN architectures. This paper extends graph diffusion to multimodal brain graphs, achieving simultaneous intra-modal and inter-modal diffusion.

**Prompt Tuning.** Prompt tuning is a fine-tuning technique that optimizes continuous prompts to improve model performance on specific tasks (Ju et al., 2022; Shen et al., 2024; Zhang et al., 2024b). Specifically, trainable continuous prompts are appended to the input representations and updated during training. Recently, prompt tuning has been applied to multimodal learning to facilitate effective and robust fusion across diverse modalities (Khattak et al., 2023; Li et al., 2024a). For example, PMF (Li et al., 2023) employs three interactive prompts, including query, query context, and fusion context prompts, to dynamically learn various objectives for multimodal fusion. In this paper, we design a prompt-based fusion strategy to enhance the robustness of structural and functional brain graph fusion.

## D    IMPLEMENTATION DETAILS

The detailed hyperparameter settings for training Xdiff on three datasets are summarized in Table 3. The model parameters are trained using the Adam optimizer. Fusion prompts $\mathbf{Z}_{\text{fp}}^{s}$ and $\mathbf{Z}_{\text{fp}}^{f}$ are obtained by random initialization and are optimized as learnable parameters during the training process. For the ADNI dataset, we set the number of diffusion layers to $K = 1$, size step to $\rho = 0.01$, fusion layer number to $L = 2$, and prompt length to $U = 4$. For the PPMI dataset, we set the number of diffusion layers to $K = 2$, size step to $\rho = 0.1$, fusion layer number to $L = 2$, and prompt length to $U = 4$. For the 4RTNI dataset, we set the number of diffusion layers to $K = 4$, size step to $\rho = 0.1$, fusion layer number to $L = 2$, and prompt length to $U = 4$.

Table 3: Hyperparameters for training on three different datasets.

| Hyperparameter | ADNI | PPMI | 4RTNI |
|---|---|---|---|
| #Diffusion layer $K$ | 1 | 2 | 4 |
| Step size $\rho$ | 0.01 | 0.1 | 0.1 |
| #Fusion layer $L$ | 2 | 2 | 2 |
| Prompt length $U$ | 4 | 4 | 4 |
| $w_1$ & $w_2$ | 0.50 | 0.50 | 0.50 |
| Dropout | 0.5 | 0.1 | 0.5 |
| Hidden channel | 64 | 64 | 64 |
| Learning rate | 5e-3 | 1e-3 | 3e-3 |
| #Epochs | 200 | 300 | 100 |
| Weight decay | 5e-4 | 5e-4 | 5e-4 |

**Number of parameters and computation time.** The number of parameters for Xdiff are 100K, 137K, and 212K on ADNI, PPMI, and 4RTNI, respectively. The running time for training Xdiff on ADNI, PPMI, and 4RTNI is 5.89 s/epoch, 4.28 s/epoch, and 5.16 s/epoch, respectively.

## E    ADDITIONAL EXPERIMENTAL RESULTS

### E.1    MODEL PERFORMANCE COMPARISON

Table 4 summarizes the experimental results (F1, Sen., and Spe.) on three datasets. The data modality used to evaluate unimodal baseline methods depends on the modality utilized in the original studies.

### E.2    RESULTS OF ABLATION STUDY

**Effectiveness of Dual Graph Diffusion.** Table 5 shows results (F1, Sen., and Spe.) of Xdiff and its variants on three datasets.

**Effectiveness of Energy Constraint Function.** As shown in Figure 4, the accuracy confidence intervals of w/o Energy on the ADNI and PPMI datasets become significantly wider compared to the original Xdiff, ranging from 67.0% to 72.2% on ADNI, and from 60.6% to 64.4% on PPMI. Results

Table 4: Experimental results (F1, Sen., and Spe.) on three datasets (%).

| Method | | ADNI | | | PPMI | | | 4RTNI | | |
|---|---|---|---|---|---|---|---|---|---|---|
| | | F1 | Sen. | Spe. | F1 | Sen. | Spe. | F1 | Sen. | Spe. |
| Unimodal | A-GCL | 34.4±7.2 | 39.9±6.0 | 71.3±3.2 | 40.8±3.6 | 48.6±8.4 | 72.1±5.5 | 41.5± 7.2 | 77.4±11.1 | 50.6±10.2 |
| | STAGIN | **51.1±2.8** | 39.6±3.5 | 71.2±2.7 | 56.3±5.2 | 46.8±8.4 | 77.2±4.3 | 69.2± 11.3 | 83.4±13.1 | 54.6±9.8 |
| | GroupBNA | 45.0±1.6 | 37.0±1.1 | 69.8±0.9 | 39.9±5.6 | 45.7±3.0 | 72.0±1.4 | 50.2± 5.9 | 72.9±3.4 | 43.8±15.8 |
| | MCST-GCN | 49.0±2.8 | 38.1±6.3 | 70.5±2.6 | 53.1±5.4 | 40.0±8.2 | 69.9±3.9 | 57.2± 20.1 | 68.2±16.4 | 47.6±13.8 |
| | NeuroPath | 49.3±2.8 | 39.6±4.3 | 72.1±5.7 | 55.7±6.2 | 46.3±7.7 | 77.4±5.3 | 67.1±10.0 | 79.7±8.2 | 51.0±8.9 |
| Multimodal | RF | 36.5±4.2 | 39.5±3.8 | 71.5±3.0 | 45.4±4.4 | 46.3±3.9 | 73.4±2.1 | 42.7±24.1 | 44.3±25.9 | **61.3±13.2** |
| | SVM | 37.9±5.3 | 41.2±4.7 | 72.8±3.7 | 31.8±2.3 | 39.9±1.4 | 70.0±0.4 | 66.6±7.6 | 63.8±12.9 | 46.1±12.3 |
| | Differmer | 46.1±6.4 | 47.1±4.9 | 75.9±2.8 | 43.3±5.5 | 43.3±5.5 | 72.4±3.6 | 49.2±12.4 | 54.0±11.2 | 53.2±11.2 |
| | DDM | 43.7±5.1 | 45.3±3.9 | 75.5±2.3 | 48.5±13.8 | 51.4±12.3 | 74.9±6.6 | 50.9±17.6 | 83.5±17.6 | 26.7±24.9 |
| | ECMGD | 42.6±4.3 | 40.4±3.1 | 74.8±2.8 | 48.1±4.3 | 49.8±3.1 | 73.3±3.2 | 62.1±4.8 | 71.5±6.6 | 52.9±5.8 |
| | Cross-GNN | 40.2±5.4 | 40.4±7.5 | 72.9±3.7 | 57.9±8.5 | 44.7±9.0 | 76.6±8.7 | 69.9±5.0 | 86.2±5.0 | 50.0±3.0 |
| | RH-BrainFS | 40.3±7.3 | 42.2±7.3 | 73.4±5.1 | 47.7±9.7 | 48.5±10.2 | 73.1±5.2 | 52.1±9.8 | 70.6±11.5 | 57.9±12.8 |
| | MTAN | 48.3±6.4 | 46.2±6.9 | 63.1±8.2 | 51.8±6.5 | **54.2±10.1** | 66.0±9.4 | 70.1±20.7 | 86.0±19.6 | 58.0±16.0 |
| | AL-NEGAT | 48.4±4.3 | 50.7±3.2 | 79.5±1.7 | 53.6±5.9 | 43.4±5.1 | 77.6±2.6 | 51.8±29.7 | 60.0±38.8 | 56.0±19.6 |
| Our Model | **Xdiff** | 49.9±0.3 | 46.8±0.4 | **81.7±0.8** | **58.8±1.9** | 45.2±0.8 | **78.0±0.1** | **77.7±0.6** | **86.9±0.5** | 53.8±5.2 |
| | $\Delta_{SOTA}$ | ↓1.2 | ↓3.9 | ↑2.2 | ↑0.8 | ↓9.0 | ↑0.4 | ↑7.6 | ↑0.7 | ↓7.5 |

Table 5: Model performance (F1, Sen., and Spe.) of Xdiff and its variants on three datasets (%).

| Method | ADNI | | | PPMI | | | 4RTNI | | |
|---|---|---|---|---|---|---|---|---|---|
| | F1 | Sen. | Spe. | F1 | Sen. | Spe. | F1 | Sen. | Spe. |
| w/o Intra | 43.7±0.4 | 41.8±0.2 | 79.6±0.1 | 43.6±0.1 | 40.4±0.4 | 77.1±0.2 | 73.8±0.4 | 83.2±1.2 | 41.6±0.9 |
| w/o Inter | 48.4±1.3 | 46.3±0.2 | 80.9±0.2 | 57.3±1.1 | 44.2±0.3 | 77.0±0.1 | 73.2±0.8 | 85.2±0.8 | 42.5±1.9 |
| w/o Intra & Inter | 43.8±0.5 | 41.4±0.4 | 79.7±0.2 | 43.6±0.1 | 40.2±0.4 | 76.8±0.2 | 74.1±0.7 | 85.1±3.2 | 52.8±1.0 |
| **Xdiff** | **49.9±0.3** | **46.8±0.4** | **81.7±0.8** | **58.8±1.9** | **45.2±0.8** | **78.0±0.1** | **77.7±0.6** | **86.9±0.5** | **53.8±5.2** |

on the 4RTNI dataset show that the accuracy confidence interval of w/o Energy is slightly wider than Xdiff. This may be attributed to the smaller sample size and higher variability in the 4RTNI dataset, which dominate the overall instability. In such cases, the energy constraint function has a limited impact on further stabilizing the diffusion process, as the primary source of instability lies in the dataset's inherent characteristics rather than the diffusion process. Overall, these experimental results demonstrate the effectiveness of the energy constraint in enhancing model stability.

**Effectiveness of Prompt-based Fusion.** Table 6, Table 7, and Table 8 gives the model performance of models using different fusion methods before and after noise interference on three datasets. As shown in Table 6, the model using the concatenation operation exhibits improved performance after noise interference on the ADNI dataset. This may be because the noise disrupts certain overfitted patterns. However, concatenation itself relies heavily on the data distribution, making it potentially unstable when faced with different datasets or varying noise levels. To further demonstrate the robustness of the PF strategy compared to the concatenation operation, we increase the noise ratio to 0.8 on ADNI and compare the performance changes. Figure 7 shows the results. "PF Before" and "PF After" indicate results of PF before and after noise interference, while "Concat Before" and "Concat After" are results of concatenation before and after noise interference. When the noise ratio is increased to 0.8, the performance of the model using concatenation fluctuates significantly, with accuracy dropping by 1.8%. In contrast, our model with PF demonstrates greater stability, with accuracy dropping by only 0.2%. This indicates that Xdiff with PF strategy is more robust compared to the model using concatenation operation.

Table 6: Performance of models using different fusion methods before and after noise interference on the ADNI dataset (%).

| Method | ADNI | | | | |
|---|---|---|---|---|---|
| | ACC | F1 | AUC | Sen. | Spe. |
| Sum. | 69.1±0.8 | 48.5±0.8 | 73.1±0.4 | 48.6±0.4 | 82.3±0.2 |
| Sum. (+Noise) | 67.5±1.5 | 47.7±0.9 | 72.0±0.5 | 47.4±0.3 | 81.1±0.2 |
| Multi. | 65.9±0.6 | 46.4±0.5 | 68.2±0.6 | 42.6±0.5 | 78.3±0.4 |
| Multi. (+Noise) | 62.6±1.2 | 44.0±1.0 | 67.6±0.6 | 42.0±0.1 | 77.7±0.1 |
| Concat. | 67.4±1.3 | 47.5±0.8 | 73.0±0.4 | 48.0±0.1 | 81.0±0.1 |
| Concat. (+Noise) | 69.1±1.0 | 48.6±0.8 | 72.6±0.6 | 47.9±0.3 | 81.8±0.3 |
| Atten. | 63.3±4.1 | 47.3±4.8 | 68.9±0.6 | 41.9±1.0 | 78.4±0.7 |
| Atten. (+Noise) | 62.1±3.2 | 47.2±3.8 | 68.4±0.7 | 40.9±0.9 | 77.9±0.8 |
| Trans. | 65.0±2.5 | 47.9±2.2 | 67.8±1.1 | 38.5±1.3 | 77.2±0.4 |
| Trans. (+Noise) | 63.4±4.8 | 47.4±1.5 | 69.1±1.2 | 40.7±0.5 | 78.0±0.3 |
| **PF** | 70.8±0.6 | 49.9±0.3 | 77.2±0.9 | 46.8±0.4 | 81.7±0.8 |
| **PF (+Noise)** | 70.8±0.6 | 49.8±0.5 | 77.1±0.2 | 46.5±0.1 | 81.6±0.1 |

Table 7: Performance of models using different fusion methods before and after noise interference on the PPMI dataset (%).

| Method | PPMI | | | | |
|---|---|---|---|---|---|
| | ACC | F1 | AUC | Sen. | Spe. |
| Sum. | 59.4±0.1 | 56.0±1.6 | 64.2±0.5 | 46.7±0.9 | 77.4±0.9 |
| Sum. (+Noise) | 56.2±2.6 | 54.9±2.9 | 62.3±0.5 | 47.8±0.5 | 76.8±0.7 |
| Multi. | 59.4±2.5 | 53.8±3.1 | 62.9±1.3 | 43.2±1.2 | 75.6±0.5 |
| Multi. (+Noise) | 52.0±3.9 | 46.6±3.4 | 61.6±0.7 | 42.2±0.3 | 75.2±0.1 |
| Concat. | 58.3±1.5 | 55.6±1.7 | 63.9±0.4 | 46.7±0.6 | 77.4±0.6 |
| Concat. (+Noise) | 55.4±1.4 | 54.1±3.3 | 61.8±0.3 | 44.6±0.6 | 77.3±0.6 |
| Atten. | 56.2±1.5 | 53.8±3.0 | 67.2±0.3 | 41.5±0.8 | 77.4±0.4 |
| Atten. (+Noise) | 52.7±2.9 | 52.5±1.3 | 66.9±0.4 | 41.7±0.8 | 76.6±0.5 |
| Trans. | 57.6±2.9 | 53.7±3.6 | 65.6±1.9 | 49.2±1.6 | 75.9±1.2 |
| Trans. (+Noise) | 55.2±3.9 | 55.8±4.5 | 65.5±1.9 | 47.0±1.7 | 75.9±1.3 |
| **PF** | 62.5±0.1 | 58.8±1.9 | 64.7±1.2 | 45.2±0.8 | 78.0±0.1 |
| **PF (+Noise)** | 60.4±1.5 | 58.3±2.6 | 64.9±0.1 | 44.1±0.3 | 77.8±0.1 |

Table 8: Performance of models using different fusion methods before and after noise interference on the 4RTNI dataset (%).

| Method | 4RTNI | | | | |
|---|---|---|---|---|---|
| | ACC | F1 | AUC | Sen. | Spe. |
| Sum. | 77.4±3.1 | 78.2±0.4 | 77.2±0.2 | 83.2±1.2 | 52.8±1.3 |
| Sum. (+Noise) | 73.6±3.1 | 74.7±0.3 | 73.0±0.6 | 82.8±0.4 | 41.6±0.9 |
| Multi. | 75.6±3.2 | 68.2±2.4 | 68.3±0.7 | 85.1±3.2 | 42.5±1.9 |
| Multi. (+Noise) | 69.3±5.4 | 68.9±3.1 | 58.4±0.8 | 87.2±3.7 | 30.7±3.3 |
| Concat. | 77.8±6.3 | 75.9±2.7 | 73.2±3.5 | 84.1±3.2 | 42.5±1.9 |
| Concat. (+Noise) | 73.3±6.3 | 75.9±2.6 | 70.2±3.6 | 82.2±3.4 | 36.9±1.8 |
| Atten. | 77.8±3.1 | 73.0±1.3 | 72.1±11.9 | 86.7±2.0 | 30.7±8.4 |
| Atten. (+Noise) | 71.1±8.3 | 72.8±2.1 | 70.4±11.7 | 84.2±1.2 | 31.9±7.8 |
| Trans. | 72.6±6.3 | 70.6±0.2 | 66.6±2.5 | 89.6±5.6 | 16.4±12.9 |
| Trans. (+Noise) | 68.9±6.3 | 69.7±1.2 | 65.7±5.9 | 89.0±5.3 | 14.9±8.3 |
| **PF** | 77.8±3.1 | 77.7±0.6 | 77.4±0.4 | 86.9±0.5 | 53.8±5.2 |
| **PF (+Noise)** | 75.6±3.0 | 77.4±0.4 | 76.0±2.7 | 85.2±1.1 | 52.8±1.3 |

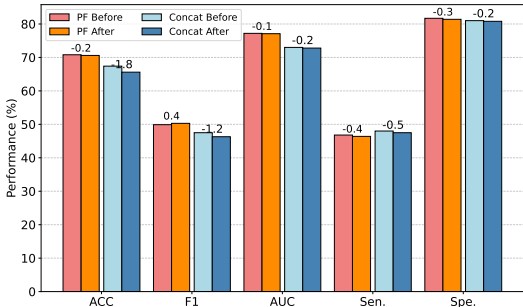

Figure 7: Model performance comparison between PF and concatenation when noise ratio is 0.8 on the ADNI dataset.

### E.3 Hyperparameter Study

Figure 8 illustrates the accuracy of Xdiff with respect to diffusion step size and diffusion layer number on the PPMI and 4RTNI datasets. Similarly, Figure 9 shows the accuracy of Xdiff with respect to fusion layer number $L$ and prompt length $U$ on the PPMI and 4RTNI datasets.

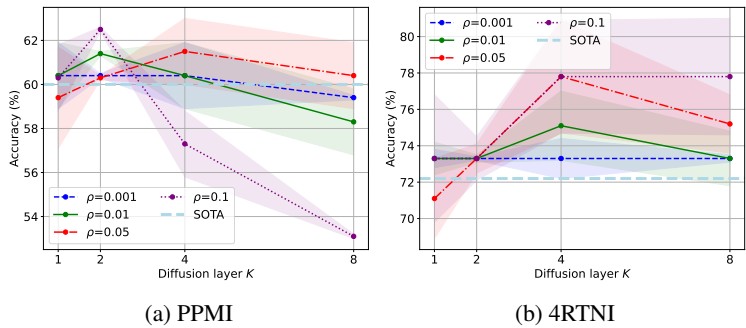

(a) PPMI      (b) 4RTNI

Figure 8: Performance of Xdiff w.r.t $\rho$ and $K$ on the PPMI and 4RTNI datasets.

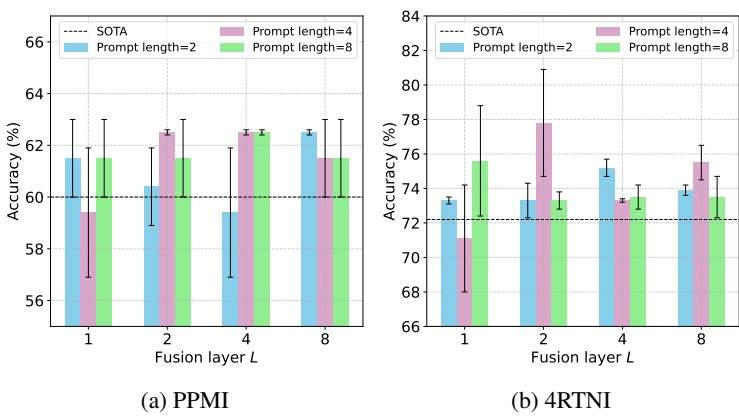

(a) PPMI      (b) 4RTNI

Figure 9: Performance of Xdiff w.r.t fusion layer number $L$ and prompt length $U$ on the PPMI and 4RTNI datasets.

**Sensitivity Analysis to Weighting Constants $w_1$ and $w_2$ in Energy Constraint.** In Equation (5), $w_1$ and $w_2$ are weighting constants, both of which are set to be $0.50$. To evaluate the sensitivity of our model on these hyperparameters, we conduct experiments by setting different values of $w_1 = (0.25, 0.50, 0.75)$ and $w_2 = (0.25, 0.50, 0.75)$. Tables 9, 10, and 11 give the results. These experiments show that our model maintains stable and competitive performance across a wide range of hyperparameter settings.

Table 9: Results (ACC) of sensitivity analysis to weighting constants $w_1$ and $w_2$ on ADNI (%).

|  | $w_1(0.25)$ | $w_1(0.50)$ | $w_1(0.75)$ |
|---|---|---|---|
| $w_2(0.25)$ | $68.5 \pm 0.3$ | $70.3 \pm 0.2$ | $70.1 \pm 0.4$ |
| $w_2(0.50)$ | $67.3 \pm 0.5$ | $\mathbf{70.8 \pm 0.6}$ | $71.4 \pm 0.3$ |
| $w_2(0.75)$ | $70.3 \pm 0.5$ | $70.5 \pm 0.7$ | $70.4 \pm 0.1$ |

Table 10: Results (ACC) of sensitivity analysis to weighting constants $w_1$ and $w_2$ on PPMI (%).

|  | $w_1(0.25)$ | $w_1(0.50)$ | $w_1(0.75)$ |
|---|---|---|---|
| $w_2(0.25)$ | $61.1 \pm 0.3$ | $60.5 \pm 0.2$ | $62.1 \pm 0.2$ |
| $w_2(0.50)$ | $62.1 \pm 0.2$ | $\mathbf{62.5 \pm 0.1}$ | $61.3 \pm 0.1$ |
| $w_2(0.75)$ | $60.8 \pm 0.3$ | $61.1 \pm 0.2$ | $61.3 \pm 0.1$ |

Table 11: Results (ACC) of sensitivity analysis to weighting constants $w_1$ and $w_2$ on 4RTNI (%).

|  | $w_1(0.25)$ | $w_1(0.50)$ | $w_1(0.75)$ |
|---|---|---|---|
| $w_2(0.25)$ | $76.2 \pm 4.2$ | $75.5 \pm 3.8$ | $78.1 \pm 4.3$ |
| $w_2(0.50)$ | $75.4 \pm 3.6$ | $\mathbf{77.8 \pm 3.1}$ | $75.9 \pm 4.3$ |
| $w_2(0.75)$ | $76.3 \pm 3.8$ | $76.1 \pm 4.2$ | $77.8 \pm 2.9$ |

### E.4 INTERPRETABILITY ANALYSIS

To assess the interpretability of the proposed method, we conduct a SHAP-based analysis to evaluate the contribution of each brain region to the classification results on all three datasets. Specifically, we compute the average SHAP values for each ROI from the learned multimodal representations from the test set. The top 15 ROIs with the highest average SHAP values for each dataset are visualized in Figure 10, Figure 11, and Figure 12.

For the ADNI dataset, regions such as the Thalamus and Fusiform are identified as highly influential and are known to be closely associated with Alzheimer's disease, in alignment with previous neuroscience findings (Forno et al., 2023; Ribeiro-dos Santos et al., 2023). In the case of the PPMI dataset, regions including the Rectus and Olfactory cortex show high importance, consistent with their established relevance to Parkinson's disease as reported in prior studies (Gu et al., 2024; Kataoka & Sugie, 2023). For the 4RTNI dataset, the Paracentral_Lobule is highlighted as a key region, which has been previously linked to Progressive Supranuclear Palsy and Corticobasal Syndrome(Kitagaki et al., 2000). These results suggest that our model can effectively identify disorder-specific biomarkers that are consistent with known neuropathological patterns, thereby demonstrating biological interpretability.

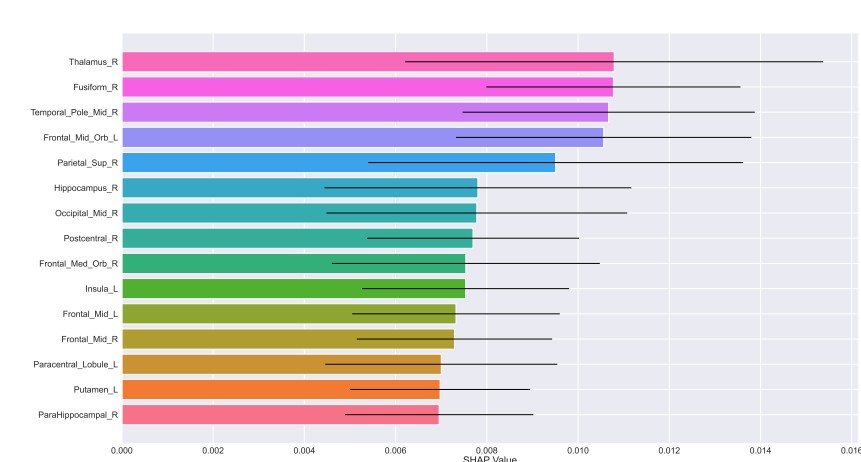

Figure 10: Visualization of the top 15 ROIs with the highest SHAP values for ADNI dataset.

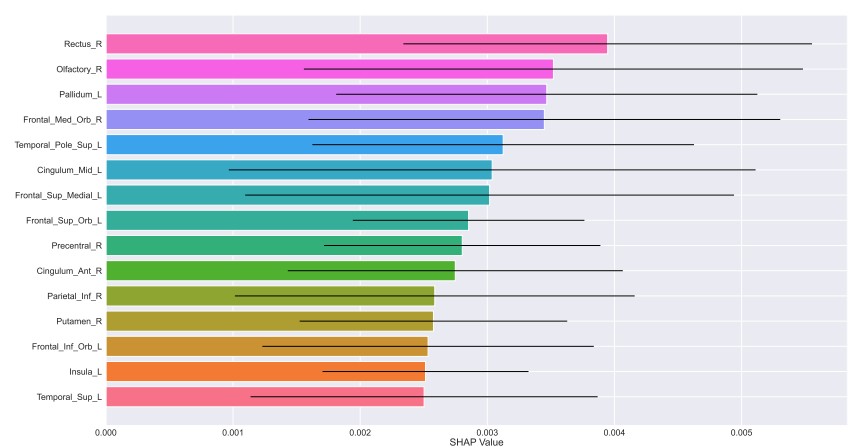

Figure 11: Visualization of the top 15 ROIs with the highest SHAP values for PPMI dataset.

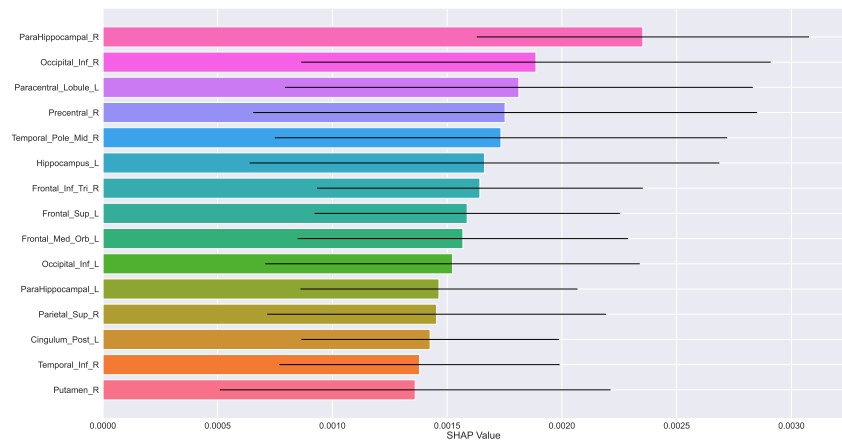

Figure 12: Visualization of the top 15 ROIs with the highest SHAP values for 4RTNI dataset.

## F  LIMITATIONS AND POTENTIAL IMPACTS

**Limitations.**    While these findings are encouraging, some limitations remain. Due to the limited available multimodal brain imaging datasets, the application of Xdiff to broader tasks is constrained. For example, the Autism Brain Imaging Data Exchange (ABIDE) dataset, one of the most commonly used datasets for autism detection, only provides fMRI data. In future work, we aim to explore more multimodal brain imaging datasets and further advance multimodal brain graph fusion for broader and more complex applications.

**Potential Impacts.**    Our work has a significant positive impact on the advancement of digital health. Importantly, this work contributes to the intersection of neuroscience and artificial intelligence by providing an effective multimodal brain graph fusion technique. However, we also acknowledge potential negative impacts. AI-based disease diagnosis may lead to erroneous predictions, which could have serious consequences for patients' health and well-being. Therefore, we emphasize that in real-world clinical settings, AI models should serve as decision-support tools, and final diagnostic decisions must be made by qualified medical professionals.

