# OpenReview forum: "Cross-modal Brain Graph Diffusion"
_ICLR.cc/2026/Conference — ICLR 2026 Conference Withdrawn Submission_

### Official Review · Reviewer_zvLc · 2025-10-26

**Soundness:** 3
**Presentation:** 3
**Contribution:** 2
**Rating:** 6
**Confidence:** 4

**Summary:**

This paper presents a novel and methodologically sound framework, Xdiff, which makes a valuable contribution to multimodal brain graph analysis. The core idea of a dual diffusion mechanism is innovative, and the experimental validation is comprehensive. However, significant concerns regarding theoretical rigor, methodological clarity, comparative fairness, and statistical depth.

**Strengths:**

1. The paper clearly identifies and addresses a critical gap in multimodal brain graph fusion: the simultaneous capture of intra-modal dependencies and inter-modal correlations. The motivation is compelling and well-grounded in neuroscientific principles.

2. The proposed dual graph diffusion mechanism is a principled and elegant extension of graph diffusion to the multimodal setting. The integration of an energy constraint for stability and a prompt-based fusion (PF) strategy for robustness represents a creative synthesis of ideas from different domains.

3. Evaluation across three distinct brain disorder datasets (ADNI, PPMI, 4RTNI) demonstrates generalizability.

**Weaknesses:**

1. The proof of Theorem 1 relies on the assumption that Sk is a stochastic matrix with λmax​(Sk)=1. The manuscript must explicitly state how this property is enforced or maintained throughout the diffusion process during training. If it is not enforced, the convergence guarantee is weakened, and an empirical analysis of the spectral properties of Sk during training should be provided to support the stability claims.

2. The inspiration and theoretical basis for the specific formulation of the energy constraint function (Eq. 5) and its connection to the diffusion process require a more detailed explanation. While Proposition 2 derives the upper bound, the initial design choice for β(⋅)and α(⋅) needs stronger motivation.

3. The feature matrix X in the Preliminaries section is not defined. The initial node features hi​(0)=xi​ for both SBG and FBG must be explicitly described. How are these initial node embeddings generated from the raw DTI/fMRI data? This is a fundamental detail currently missing.

4. Figure 2, in its current form, is insufficient. It omits critical elements like the input adjacency matrices (As,Af), the internal computation of inter-modal coefficients (c_i^s,c_i^f​) via Q/K, and the role of prompts in the fusion module. A redesigned figure or the addition of a high-level algorithm pseudocode is strongly recommended to elucidate the end-to-end data flow.

5. The description of the prompt-based fusion strategy is vague. The prompts Z_fp^s​ and Z_fp^f are stated to be "defined," but it is unclear if they are randomly initialized learnable parameters or derived from another process. This must be clarified in Section 4.3.

6. Given that SBG and FBG node features originate from fundamentally different distributions (DTI vs. fMRI), a brief discussion on whether any feature projection/normalization is applied to align the modalities before diffusion would be beneficial.

7. Reporting means and standard deviations over 5 runs is a start, but it is insufficient for small-sample datasets like PPMI and 4RTNI. Statistical significance tests (e.g., paired t-tests or non-parametric equivalents with appropriate correction for multiple comparisons) must be performed to confirm that the performance improvements over baselines are not due to chance.

8. The claim that all baselines use "optimal configurations" is not verifiable. The authors must provide a detailed account of the hyperparameter tuning process for all baselines, ensuring a fair comparison. Were the same data splits and random seeds used? Was the search space comparable? This is critical for reproducibility.

9. The manuscript should discuss cases where Xdiff does not achieve the top performance on all metrics (e.g., AUC on PPMI is lower than STAGIN, Sen. on ADNI is lower than AL-NEGAT). A thoughtful analysis of the potential reasons (e.g., model biases, dataset characteristics) would provide a more balanced perspective.

10. The "Related Work" section should be reorganized to more sharply critique existing methods, clearly separating which ones fail to capture intra-modal dependencies and which ones fail to capture inter-modal correlations. This will more effectively frame the specific niche Xdiff occupies.

11. An analysis or discussion of the model's computational cost and scalability relative to the baselines is missing and would be valuable for practitioners.

**Questions:**

Please see Weaknesses.

---

### Official Review · Reviewer_hNYQ · 2025-10-28

**Soundness:** 2
**Presentation:** 2
**Contribution:** 1
**Rating:** 2
**Confidence:** 5

**Summary:**

The paper introduces a novel method called Cross-modal Brain Graph Diffusion (Xdiff) for multimodal brain graph fusion, aiming to improve brain disorder detection. Xdiff employs a dual graph diffusion mechanism that captures both intra-modal dependencies and inter-modal correlations through separate diffusion processes for structural and functional brain graphs. Additionally, it uses an energy constraint function to ensure consistent feature propagation and enhance model stability. A prompt-based fusion strategy is also proposed to integrate multimodal features more effectively. Experiments on three datasets (ADNI, PPMI, and 4RTNI) demonstrate that Xdiff outperforms state-of-the-art methods in accuracy and stability.

**Strengths:**

1. The paper deals with a significant topic in neuroscience. It presents an advancement in multimodal brain graph fusion. The proposed Xdiff approach effectively captures both intra-modal dependencies and inter-modal correlations, leading to improved performance and stability in brain disorder detection tasks.
2. Results on comparison experiments seem good, which shows Xdiff’s effectiveness. Ablation results seem convincing to further prove modules’ effectiveness.

**Weaknesses:**

This paper has major issues in terms of research motivation, methodological innovation, and experimental setup. Please refer to the Questions section for details.

**Questions:**

1. Novelty: Your proposed Xdiff shows a high degree of overlap with ECMGD (see [1]) in terms of core innovation, theoretical analysis, and technical methodology. Since ECMGD is also concerned with multi-view graph fusion, the originality of this paper is limited. The authors need to explicitly clarify the differences between Xdiff and ECMGD.
2. Motivation claim: In Introduction part, the author claimed that current methods failed to capture intra-modal dependencies and inter-modal correlations. However, capturing intra and inter modal features is not uncommon in multi-modal brain fusion.
3. Experiments shortcomings: Experiments are conducted only on ADNI, PPMI and 4RTNI datasets. However, dataset like HCP which involves over 1000 samples is also a public mutli-modal graph dataset which is common in multi-modal network analysis. Furthermore, diffusion-based methods require great GPU resources for training, and experiments on training resources statistics and training/inference speed is required.

[1] Lu J, Wu Z, Chen Z, et al. Towards Multi-view Consistent Graph Diffusion[C]//Proceedings of the 32nd ACM International Conference on Multimedia. 2024: 186-195.

---

### Official Review · Reviewer_t8Nn · 2025-10-31

**Soundness:** 3
**Presentation:** 2
**Contribution:** 2
**Rating:** 6
**Confidence:** 3

**Summary:**

The paper proposes Xdiff (Cross-modal Brain Graph Diffusion) for fusing structural (DTI-derived SBGs) and functional (fMRI-derived FBGs) brain graphs. The key idea is a dual graph diffusion with (i) intra-modal diffusion to preserve modality-specific topology and (ii) inter-modal diffusion to model structural–functional coupling. Training uses an explicit-Euler update with a theoretically bounded step size and adds an energy constraint to encourage consistent diffusion and improve stability; finally a prompt-based fusion (PF) module integrates modalities for classification. Experiments on ADNI, PPMI, and 4RTNI show state-of-the-art accuracy.

**Strengths:**

1. The dual (intra/inter) diffusion explicitly targets the two essential attributes of multimodal brain-graph fusion (topology preservation and cross-modal coupling).
2. The update rule is spelled out, and the paper proves a concise convergence condition \(0<\rho<1\).
3.  The **energy constraint** formalizes intra- and inter-modal consistency, with a variational upper bound and conditions on the induced coefficients.

**Weaknesses:**

- Generalization is under-specified: multi-site confounds (site/batch, motion, age/sex) and held-out-site validation are not detailed, which is important for correlation-based graphs.
- The energy term relies on unspecified non-decreasing concave functions \(\alpha,\beta\) and weights \(w_1,w_2\); concrete choices and ablations are limited in the main text.
- Theory assumptions may not automatically hold during training for data-driven inter-modal coefficients; no spectral diagnostics are reported.
- Reported improvements emphasize accuracy; threshold-free metrics (e.g., AUC) are not uniformly dominant across datasets.

**Questions:**

See above

---

### Official Review · Reviewer_th4r · 2025-10-31

**Soundness:** 3
**Presentation:** 3
**Contribution:** 2
**Rating:** 4
**Confidence:** 4

**Summary:**

This paper introduces Xdiff, a novel framework for learning from multimodal brain graphs (fMRI and DTI).
It proposes a dual graph diffusion process to jointly model intra-modal (within-modality) and inter-modal (cross-modality) dependencies, combined with an energy constraint to maintain stability and a prompt-based fusion (PF) module for adaptive feature integration.
Experiments on ADNI, PPMI, and 4RTNI datasets show moderate improvements over several baselines for disease classification (Alzheimer’s, Parkinson’s, tauopathy).

**Strengths:**

The paper presents an interesting and technically well-motivated framework for multimodal brain graph learning. Its dual graph diffusion mechanism and energy constraint are conceptually novel and mathematically sound, and the idea of incorporating a prompt-based fusion strategy shows creativity and cross-domain thinking. The theoretical grounding and stable training behavior are commendable, and the experiments demonstrate that the approach is feasible and generally consistent across multiple datasets. Overall, the work reflects solid effort and technical competence.

**Weaknesses:**

The empirical results are underwhelming — the reported accuracies (around 70% on ADNI and 62% on PPMI) are lower or comparable to prior works. The performance improvements (2–6%) are minor and within typical experimental variance. Moreover, the method introduces substantial computational complexity without a clear analysis of runtime or scalability. The prompt-based fusion component feels heuristic and insufficiently justified for neuroimaging data. Evaluation is limited to small, disease-specific datasets, with no cross-site or cross-dataset validation to support generalization. The work also lacks biological interpretability, an essential aspect in medical AI applications.

**Questions:**

1. Conceptually novel but empirically modest – The proposed dual graph diffusion and energy-constrained design are innovative, but the reported improvements over prior multimodal graph methods are small and do not convincingly establish a performance advantage.

2. Complex model with limited analysis – The framework introduces several interacting modules (dual diffusion, energy constraint, and prompt-based fusion), yet the paper lacks discussion of parameter sensitivity, hyperparameter effects, or computational cost, which makes it difficult to assess the robustness and practicality of the approach.

3. Weak generalization and interpretability – The evaluation is restricted to small, disease-specific datasets without cross-site validation, and the paper offers little insight into what the model learns or how the diffusion and fusion mechanisms relate to underlying brain structures or functions.

---

### Note · Authors · 2025-11-12

I have read and agree with the venue's withdrawal policy on behalf of myself and my co-authors.